

# Understanding epistemic uncertainty in large-scale coastal flood risk assessment for present and future climates

Michalis I. Vousdoukas[1,3], Dimitrios Bouziotas[1,2], Alessio Giardino[2], Laurens M. Bouwer[2], Evangelos Voukouvalas[1], Lorenzo Mentaschi[1] and Luc Feyen[1]

[1] European Commission, Joint European Research Centre (JRC), Via Enrico Fermi 2749, I-21027-Ispra, Italy
[2] Deltares, P.O. Box 177, 2600 MH Delft, the Netherlands
[3] Department of Marine Sciences, University of the Aegean, University hill, 41100, Mitilene, Lesbos, Greece

*Correspondence to*: Michalis I. Vousdoukas (Michail.VOUSDOUKAS@ec.europa.eu)

**Abstract.** An upscaling of flood risk assessment frameworks beyond regional and national scales has taken place during recent years, with a number of large-scale models emerging as tools for hotspot identification, support for international policy-making and harmonization of climate change adaptation strategies. There is, however, limited insight on the scaling effects and structural limitations of flood risk models and, therefore, the underlying uncertainty. In light of this, we examine key sources of epistemic uncertainty in the Coastal Flood Risk (CFR) modelling chain: (i) the inclusion and interaction of different hydraulic components leading to extreme sea-level (ESL); (ii) inundation modelling; (iii) the underlying uncertainty in the Digital Elevation Model (DEM); (iv) flood defence information; (v) the assumptions behind the use of depth-damage functions that express vulnerability; and (vi) different climate change projections. The impact of these uncertainties to estimated Expected Annual Damage (EAD) for present and future climates is evaluated in a dual case study in Faro, Portugal and in the Iberian Peninsula. The ranking of the uncertainty factors varies among the different case studies, baseline CFR estimates, as well as their absolute/relative changes. We find that uncertainty from ESL contributions, and in particular the way waves are treated, can be higher than the uncertainty of the two greenhouse gas emission projections and six climate models that are used. Of comparable importance is the quality of information on coastal protection levels and DEM information. In the absence of large-extent datasets with sufficient resolution and accuracy the latter two factors are the main bottlenecks in terms of large-scale CFR assessment quality.

## 1. Introduction

Large-scale flood risk assessments have emerged during the last decade, with multiple continental or global studies focusing on fluvial risks (Winsemius et al., 2016;Alfieri et al., 2017;Alfieri et al., 2016;Dottori et al., 2017;Dottori et al., 2016b), and fewer on coastal losses (Hinkel et al., 2014;Hallegatte et al., 2013;Vousdoukas et al., 2018a). The emergence of these assessments comes as a response to the growing demand for up-scaled flood risk estimation under present and future scenarios. Such analyses can support among others adaptation planning, policy-making and applied engineering activities. Despite the eminent usefulness and promising nature of large-scale flood risk modelling frameworks, they are characterized





by a certain degree of simplification, which is the result of methodological limitations, computational restrictions, as well as lack of consistent datasets across these scales. All the above introduce significant uncertainties, affecting the accuracy of the findings.

Of the large-scale fluvial frameworks, notable discussions on model uncertainty appear in the works of Winsemius et al.

(2013), who discuss in detail, mainly qualitatively, aspects of climate input and hydrological uncertainty, as well as Ward et al. (2013), who include a sensitivity analysis to the climatic input and the assumed flood protection standards. Other studies discuss uncertainties related to the extreme value analysis (Wahl et al., 2017;Apel et al., 2008). Some coastal studies discuss the effect of DEM corrections for spatial variations of the mean sea level (Muis et al., 2017), and the inundation modelling approach (Vousdoukas et al., 2016;Breilh et al., 2013;Seenath et al., 2016). However, still many of the above factors remain

not fully quantified, especially in a unified framework which would allow assessing their relative importance.

Flood risk estimation exhibits methodological differences depending on the scale of application (Apel et al., 2009;Ferreira et al., 2017;de Moel et al., 2015). Local studies benefit from high quality digital elevation models (DEMs) (Vousdoukas et al., 2012) and exposure data (Christie et al., 2017), as well as detailed numerical models, resolving several scales (Bertin et al., 2014;Giardino et al., 2018), and including complex processes like dune or dike breaching (Plomaritis et al., 2017;de Moel et

al., 2012). Large-scale assessments, on the other hand, are based on lower resolution DEMs and exposure data and more abstract conceptualizations of physical processes: principally the hydrology and hydraulics, as well as flood damage processes. In addition, vulnerability information is often limited. At best quantitative data exist in the form of depth-damage curves (Ward et al., 2013;Hallegatte et al., 2013), but often they are omitted with some studies estimating potential exposure instead of actual quantitative risks (Neumann et al., 2015;Jongman et al., 2012b). Data scarcity is also apparent in the

assumptions behind flood management and adaptation options, commonly expressed as flood protection levels. Efforts to present consistent flood protection information for large scales have appeared only recently and are limited to fluvial cases (Scussolini et al., 2015). In coastal settings, information on flood protection remains sparse and heterogeneous, despite recent contributions in multi-case data collection (Jonkman et al., 2013;Vousdoukas et al., 2018a).

In light of this background, we develop an analytical framework that treats multiple sources of epistemic uncertainty within a

25 large-scale Coastal Flood Risk (CFR), LISCOAST: a modular, integrated coastal flood risk assessment framework developed by the Joint Research Centre of the European Commission (Vousdoukas et al., 2018a). The analysed factors include (i) the components contributing to Extreme Sea Levels (ESLs), presently focusing on waves and tides; (ii) different algorithms for inundation mapping; (iii) digital elevation data (DEM); (iv) coastal flood protection information; (v) vulnerability assumptions; and (vi) different climate change projections. For each one of these sources, variability in CFR

estimates is tested through sensitivity analyses among different conceptualizations, ranges of variables and different datasets. The analytical framework is applied for both baseline climate and under future concentration pathways, allowing to assess model uncertainty propagation in future CFR projections.





## 2.  Case Studies

The developed framework is applied in a dual case study that spans across two spatial scales and consists of: (i) a local application in Ria Formosa, Algarve, Portugal, a coastal stretch of approx. 36 km; and (ii) a regional application along the Iberian Peninsula coastline, including Portugal and Spain, with a total coastal length of 6767 km (Fig. 1 and Table 1).

Iberia, is selected as an interesting macro-scale study, featuring an extended coastline, with varying environments, like the wave and tidally dominated northeast Atlantic and the micro-tidal, less energetic Mediterranean Sea. The coastline features extensive tourist and recreational uses and a large number of exposed assets, contributing around 10% of the total flood damage for Europe (Vousdoukas et al., 2018a).

The local case study consists of the tidal lagoon and barrier island system of Ria Formosa in Faro, Portugal. It combines
oceanic waves and a tidal range up to 3 m (Vousdoukas, 2014). Apart from the complex topography (Vousdoukas, 2012), it faces socio-economic challenges, with dense infrastructure and conflicting uses, such as an airport, tourist and wildlife areas. Ria Formosa also provides a test-bed for sensitivity analysis due to the availability of a variety of in-situ measurements, including high-resolution topographic data from a LIDAR survey.

To optimise the calculations, the study areas are divided in segments with a length of 25 km along the coastline. This results
in 6 segments for Ria Formosa and over 700 segments for the Iberian Peninsula (Fig. 1, Table 1).

## 3.  Data and Methods

### 3.1  The LISCOAST methodology

The present analysis is based on the CFR model LISCOAST (Large-scale Integrated Sea-level and COastal ASsessment Tool; Fig. 2). The modular framework aims to assess weather-related impacts in coastal areas in present and future climates
by combining state-of-the-art large-scale modelling and datasets of hazard, exposure, and vulnerability (Fig. 1) (Vousdoukas et al., 2018a). The present application focusses on direct, tangible losses from flooding by sea water, which typically dominates total impacts when expressed in economic terms.

### 3.1.1  Extreme sea levels

Coastal flood impacts are driven by nearshore ESLs. In this study they are available every 25 km along the European
coastline, and ten years during the present century, for Representative Concentration Pathways scenarios RCP4.5 and RCP8.5 as simulated by 6 climate models (see Table 2), and for eight different return periods between 2 and 1000 years (Vousdoukas et al., 2017). ESLs are calculated by adding linearly the contributions of different components:

$$ESL = SLR + \eta_{CE} + \eta_{tide} \tag{1}$$

where:





- SLR is the Sea Level Rise, obtained from a GCM ensemble combined with contributions from ice-sheets and ice-caps (Vousdoukas et al., 2017).

- $\eta_{CE}$ is the contribution from extreme wind and atmospheric pressure, driving waves and storm surge, and is obtained for present and future scenarios from dynamic ocean simulations (Mentaschi et al., 2017;Vousdoukas et al., 2017) and

are available for the specified return periods $T$, after non-stationary extreme value analysis (Mentaschi et al., 2016). Given that waves are often omitted in large-scale impact assessments we evaluate the resulting error from this assumption (see Section 3.2.1).

- $\eta_{tide}$ the maximum tidal level .

### 3.1.2 Coastal inundation

ESL are used as forcing for the inundation calculations at 100 m resolution and are based on land surface elevation data provided from the Shuttle Radar Topography Mission (SRTM) DEM (Reuter et al., 2007). The inundation calculations are limited to 50 km from the coastline. For the present study the following algorithms are considered (Vousdoukas et al., 2016):

- A static, "bath-tub" approach ($S_{NH}$), where the inundation water depth in every raster cell is computed as the difference between the terrain elevation and the forcing nearshore water level ESL (Apel et al., 2009):

- A static approach that also considers hydraulic connectivity ($S_H$) with adjacent cells of which the elevation is lower than ESL.

- A quasi-dynamic algorithm that takes into account the volume of water available for inundation (VI). This approach is presented as '$SO$' in Breilh et al. (2013) and assumes a design hydrograph driven by meteorological factors, that is added to the high tide water level to calculate the time-dependent total water level. The design storm surge hydrograph

requires information about the extreme event duration $D_{CE}$. This is obtained after analysing the hydrographs of all extreme events at each coastal point and correlating $D_{CE}$ with the peak $\eta_{CE}$ through a linear function (Vousdoukas et al., 2016). ESL time series can thus be converted in flow discharge and whenever the ESL exceeds the flood protection elevation, inundation initiates. Since the flood duration is limited by the hydrograph, so is the effective flood water volume $VI_{effective}$. The final step is to apply the $S_H$ method but increasing incrementally the forcing water level until the

inundation volume is equal to $VI_{effective}$.

### 3.1.3 Impact assessment

The resulting inundation maps are combined with exposure and vulnerability information to estimate direct flood damages (Vousdoukas et al., 2018a). Exposure is available from the refined CORINE land use/land cover dataset (CLC) at 100 m resolution, featuring 44 different land use classes (Batista e Silva et al., 2012). The vulnerability to coastal flooding of

coastal infrastructure, societies and ecosystems is expressed through depth-damage functions (Rojas et al., 2013;Alfieri et al., 2015). DDFs define for each of the 44 land use classes of the refined CORINE Land Cover the relation between flood inundation depth and direct damage. The country-specific DDFs were further rescaled at NUTS3 level based on 2010 GDP





per capita to account for differences in the spatial distribution of wealth within countries. Areas that lie below the high tide water level are considered as fully damaged and the maximum loss according to the DDFs is applied. For areas inundated during extreme events, the damage is estimated by applying the DDFs combined with the simulated inundation depth for the respective events.

The impact assessment was performed at 100 m spatial resolution and the year 2010 is considered as reference year with consequent time steps of 10 years until the end of the century. For each point in time the potential impacts are first estimated for each of the considered return periods. The Expected Annual Damage (EAD) is then estimated by integrating the resulting damage probability curves.

## 3.2   Exploring uncertainties

### 3.2.1   Tidal contributions to ESLs

Eq. (1) uses a single value for the tidal contribution to ESLs, i.e. equal to the maximum tidal amplitude $\eta_{tide}$. This assumes that all extreme weather events coincide with the highest possible tide, thus ignoring spring-neap tidal variability. To investigate the limitations of this modelling assumption, a $\eta_{tide}$ modulation factor $\alpha$ is introduced in Eq. (1):

$$ESL = SLR + \eta_{CE} + \alpha \cdot \eta_{tide} \qquad (4)$$

Given that extreme events normally last several hours, they coincide at least once with high tide, the height of which depends on spring-neap tidal variability. The valid range of $\alpha$ was estimated after exploring possible combinations of extreme events with tidal signals in a stochastic way through the following steps:

- historical tide gauge records obtained from the UHSLC global tide gauge database (http://uhslc.soest.hawaii.edu) were used to provide time series of tidal elevations from one tidal gauge in Portugal and two in Spain; one in the

Mediterranean and one in the Atlantic (see Fig. 1a). An annual slice with the lowest possible number of missing values (less than 3% of the total tidal record) is then extracted and used in the following analysis.

- stochastic $\eta_{CE}$ hydrographs as described in Section 3.1.2 are super-imposed on the obtained tidal signals, based on a preset seasonal distribution typical of European coastal storms (Menéndez and Woodworth, 2010;see also approach of Lozano et al., 2004). This superimposition is considered a random event, assuming that the starting hour of the storm

$t_{start}$ is a random variable within the annual duration of the tidal signal that follows the aforementioned seasonal distribution. For each synthetic $\eta_{CE}$ event, the maximum tidal amplitude that is observed during this event is isolated – as representative of the worst-case $\eta_{CE}+\eta_{tide}$ combination and the factor $\alpha$ is calculated. A sample size of $10^4$ events is chosen and a Monte-Carlo analysis is performed, leading to the empirical distribution of the $\alpha$-factor, from which the mean value E($\alpha$) is retained as a representative single estimate.





The above analysis showed that a valid range for $\alpha$ within $0.5<\alpha<1$. In order to estimate how the variability of $\alpha$ affects the estimated losses we conducted the impact analysis increasing the parameter with an increment of $d\alpha=0.1$.

### 3.2.2     Wave contributions to ESLs

Omitting contributions from waves to ESLs is a common abstraction in large-scale CFR assessments, even though wave contributions can be important depending on the nearshore wave climate (Serafin and Ruggiero, 2014;Vousdoukas et al., 2016;Melet et al., 2018). To investigate the effects of this omission we consider the wave setup contributions to ESLs. These are estimated using the approximation of $\eta_{wave}=0.2H_s$ (Camfield, 1991), with $H_s$ being the significant offshore wave height, available from a global wave reanalysis (Mentaschi et al., 2017). The wave contribution is then added to the storm surge levels to produce new $\eta_{CE}$ values contributing to ESLs through Eq. (1).

### 3.2.3     Inundation algorithms

This section relates to flood inundation modelling methodological simplifications and underlying assumptions detailed in Section 3.1.2. For the 3 different approaches presented ($S_H$, $S_{NH}$ and VI) inundation maps are derived and used to estimate and compare EADs.

### 3.2.4     Digital Elevation Model

Global DEMs like the 100 m SRTM affect the quality of large scale assessments by (i) simplifying the terrains relief; (ii) adding systematic bias; and (iii) not resolving natural or artificial coastal protection elements. To appraise the above uncertainties we use high-quality 0.5m resolution LIDAR nearshore elevation data available for Faro Beach (Vousdoukas et al., 2012). In order to quantify the effect of DEM resolution on CFR assessment we create 4 alternative DEMs by resampling the LIDAR dataset in 10, 20, 50 and 100 m resolution and we compare against the SRTM DEM both in terms of vertical elevation as well as the resulting EADs. Given that the computation cost of the inundation analysis increases exponentially with DEM resolution, in contrast with the other uncertainty factors the comparison is restricted only to the median baseline scenario.

### 3.2.5     Coastal flood protection

Global DEMs lack the resolution to resolve coastal protection elements (see also section 4.1), which is often treated as a sub-grid process and is explicitly parameterized either in the inundation (Vousdoukas et al., 2016), or in the impact assessment module (Alfieri et al., 2017). As usually the case in large-scale impact assessments (Scussolini et al., 2015), a uniform crest level $z_{crest}$ is considered along each coastal segment. Consequently, flooding is activated only when $z_{crest}$ is exceeded by the forcing ESLs. Given that protection information is scarce and when available comes with low detail and accuracy, it is an important source of uncertainty. Therefore, a sensitivity analysis is performed increasing $z_{crest}$ within a range from 0.0 m to 2.0 m, with an increment of dz=0.5 m and the resulting EADs are compared. Similarly to the tidal elevation uncertainty



analysis (see Section 3.2.1), the range of the applied $z_{crest}$ perturbation was based on the observed errors of reported flood protection levels against in situ measurements.

### 3.2.6    Vulnerability

Vulnerability is expressed through DDFs (see section 3.1.3) that were initially derived for fluvial flood risk (referred to as

$DDF_L$) estimation and, as a result, do not account for factors such as wave forces and salinity. The choice of the DDFs is justified by the fact that they have been calibrated and validated at pan-European scale with satisfactory results (Jongman et al., 2012a). We formulate an alternative set of DDFs ($DDF_A$) based on a number of smaller scale coastal studies (Table 3). Among the 5 main land use categories of $DDF_L$, we have compiled and produced updated $DDF_A$ information for 4 (*residential, commercial, industrial, and agricultural*), while for *infrastructure* no new DDFs could be derived due to a lack

of data. We apply the same contribution of the main land use categories to the different CLC land use classes as for $DDF_L$ to arrive to the updated $DDF_A$ for each CLC land use class.

   $DDF_A$ have a sharper concave form compared to $DDF_L$, leading to a higher damage percentage for smaller depths (Fig. 3). Both vulnerability datasets are used to perform comparative runs in the studied cases and the resulting EADs are compared.

### 3.2.7    Assessing the relative importance of the uncertainty factors

We consider the following setup as the *Default* one: ESLs considering the maximum tide and no waves, inundation maps estimated with VI, DEM derived from 100 m SRTM, flood protection from FLOPROS and standard LISCOAST DDFs ($DDF_L$). Then we assess how varying each uncertainty factor separately affects the amplitude and temporal evolution of the estimated EAD for each study area. In order to focus only on the effect of the factor studied we average the median case from each RCP studied. In addition, we estimate the *very likely* range ($5^{th}$-$95^{th}$ quantile) of the *Default* setup for each RCP to

obtain an estimate of the uncertainty related to future greenhouse gas emissions and climate prediction.

   To gain further insight on the relative importance of each uncertainty factor, we first consider only results for the baseline period. Varying one parameter at the time we create groups of EAD estimates.  The deviation of the median EAD of the group from the *Default* setup EAD expresses the effect of the uncertainty factor to the estimated losses, while the range of the EAD values expresses the introduced uncertainty. In addition we create similar groups but only for both the absolute

($\Delta$EAD) and percentage change ($\Delta$EAD%) towards the end of the century. The range of each group is considered as a proxy of the uncertainty from each factor.

## 4.    Results

### 4.1    Digital Elevation Model

Considered the LIDAR DEM as ground truth, we assess the accuracy of the SRTM dataset along Ria Formosa (Fig. 4).

Subsampling the LIDAR dataset at 100 m resolution we find average vertical BIAS of 1.20 m and RSME of 2.15 m for




SRTM. Such error is significant for the scope of the study, but  is lower compared to previously reported estimates (Rodríguez et al., 2006), since SRTM accuracy has improved since then. An important artefact introduced by the SRTM relates to the fact that the 100 m resolution does not resolve the dune profile, therefore the coastal protection in the study area is underestimated. This is similar for DEMs generated after sub-sampling the LIDAR dataset; in the case of Ria

Formosa a resolution of minimum 20 m is needed to resolve the dune structure (Appendix; Figure A1). This highlights that for CFR studies considering such coarse resolutions, coastal protection should be dealt with as a sub-grid process that needs parameterization.

Results for DEM of different resolutions confirms that the latter's accuracy and abstraction affect substantially the estimated losses. In the case of Ria Formosa, reducing the DEM resolution appears to result in higher losses. However, this can be a

10 site-specific effect of the local topography and demands further research before drawing more general conclusions. The EAD from the 100 m LIDAR DEM is more than double than the one from the SRTM, and almost triple than the 10 m LIDAR DEM (with parameterized coastal protection) (Fig. 4c). It is noteworthy that the latter is comparable with the EAD from the SRTM, but this is only due to the site-specific calibration of the coastal protection based on previous studies (Vousdoukas et al., 2012). For most areas such datasets are not available and deviations in the estimated losses can be substantially higher.

## 4.2     Coastal protection

As expected, raising the flood defences reduces the estimated EAD (Fig. 5a, Fig. 6a). However, considering future CFR, the effect of higher protection on the projected EAD is non-linear, especially in the case of Ria Formosa (Fig. 5a). Additional 0.5 m of protection ($\Delta z_{\text{protection}}$) does not have any risk reduction effect. This is due to the low protection standards in place, as the area is known to experience damages almost annually (Almeida et al., 2011a;Almeida et al., 2011b), while the most frequent

event analysed here is with return period of 5-years. $\Delta z_{\text{protection}}$=1 m results in lower EAD, however after 2040 the damages tend to converge towards the *Default* case, becoming equal after 2070. Apparently this 'saturation' is a combined result of the small geographic extent of the Ria Formosa site, and which therefore can be rather easily completely flooded, and the low-lying terrain. The case of $\Delta z_{\text{protection}}$=1.5 m is similar to $\Delta z_{\text{protection}}$=1 m with the difference that the initial EAD reduction is much higher and the convergence with the *Default* case takes place only towards the end of the century. Finally, with

coastal defences upgraded by 2 m, the EAD remains until 2050 below baseline levels of the default case.

In the Iberian Peninsula additional protection appears to drive in all cases incremental increases in baseline and future EAD (Fig. 6a). Diversified behaviour is observed mainly for $\Delta z_{\text{protection}}$=1 m, the EAD of which is more similar to the one of $\Delta z_{\text{protection}}$=0.5 m around the baseline, and gradually converges towards the one of $\Delta z_{\text{protection}}$=1.5 m. As a result of the above, the estimated damage reduction is higher compared to the Ria Formosa case, and especially towards the end of the century

$\Delta z_{\text{protection}}$=2 m results in a 60% EAD reduction (30% in Ria Formosa). However the EAD increase is projected to accelerate at the Iberian Peninsula, and even 2 m higher coastal defences are not sufficient to maintain the EAD below baseline levels after 2030.





### 4.3 ESL contributions

In agreement to previous findings (Vousdoukas et al., 2016) our analysis shows that omitting wave contributions to ESLs results in substantial EAD underestimation (Fig. 5b and Fig. 6b). The baseline values can almost double after including wave setup. Also the increase rate is higher, yet the relative importance of the waves reduces with time due to the increasing
dominance of SLR in the total flood damage. Considering spring-neap tidal variability through different α-factors (see Section 3.2.1) tends to reduce EADs, especially at Ria Formosa where $\alpha=0.5$ results in a 66% EAD reduction throughout the century. The tidal modulation effect is weaker at the Iberian Peninsula, reducing EAD by around 33%. This is due to the fact that a significant part of the latter consists of micro-tidal environments.

### 4.4 Inundation algorithms

In both case studies the simplest static approach $S_{NH}$ results in higher EAD, especially at the Iberian Peninsula where the differences vary within 25% and 45% (Fig. 6d). For the same site, adding hydrological connectivity ($S_H$) results in lower EAD, which decreases further when the effective water volume is considered (VI). At Ria Formosa, using $S_H$ and VI result in equal values (Fig. 5d), again a result of the restricted domain, which means that the effective flood water volume estimated by VI is still sufficient to flood the entire area. In both sites the EAD differences between the two approaches are small
(<5%), while $S_{NH}$ deviates substantially. This is due to the fact that the Iberian coastline is steep, while the static approach tends to overestimate flood extents to a larger extent in mildly sloped terrains (Vousdoukas et al., 2016).

### 4.5 Vulnerability

The alternative DDFs show higher impacts for lower inundation depths compared to the default ones (Fig. 3), resulting in slightly higher EADs for both sites (Fig. 5e and Fig. 6e). The effect is more prominent at Ria Formosa where differences are
within the 10%-15% range, compared to the Iberian Peninsula (<10%). Overall switching between the 2 tested DDFs appears to have a small effect on the estimated losses in the two case studies.

### 4.6 Relative importance of uncertainty factors

The comparison of the baseline EADs obtained from the analysis of each studied factor highlights that omitting wave contributions in ESLs is the strongest source of epistemic uncertainty at both studied cases (Fig. 7a,b). In addition, baseline
EAD at the Iberian Peninsula appears to be also sensitive to coastal protection, the inundation approach applied, followed by tidal modulation (Fig. 7b). The local analysis at Ria Formosa shows also that DEM errors can introduce substantial uncertainty (Fig. 7a).

Relative contributions in projected EAD changes vary depending on whether absolute (Fig. 7c-d), or relative changes are considered (Fig. 7e-f). The reason is that a factor which can increase both the baseline and future values (i.e. considering
waves) may result in higher absolute ΔEAD but lower ΔEAD%, due to the fact that in the latter the denominator is higher.



Considering absolute changes, factors affecting ESLs (waves, tides) are the main source of uncertainty in Ria Formosa, almost comparable to the climate change uncertainty (Fig. 7c, g). Greenhouse gas emission uncertainty comes as the fourth ranked factor. When relative contributions are considered, climate projection uncertainty becomes prominent, followed by flood protection and tidal modulation. The epistemic uncertainty from including waves is similar to the one resulting from the greenhouse gas emission scenarios. The absolute contributions in the Iberian Peninsula are more balanced, with the following order: waves, flood protection, inundation algorithms, climate projection uncertainty and tidal modulation (Fig. 7d, h). Again the uncertainty factor ranking changes when relative changes are considered; i.e. flood protection, climate uncertainty, greenhouse gas emission, and tidal modulation.

## 5. Discussion

The present analysis, while perhaps not exhaustive, provides a very useful indication of uncertainties large-scale coastal flood risk modelling, and points to the challenges of gathering sufficiently reliable data. An important conclusion is also that the relative contributions of the uncertainty factors are not generally valid, but depend on site-specific conditions, available data and methods used as shown here through the two case studies. Also the considered range of the studied parameters (e.g. $\alpha$, $\Delta z_{protection}$) has a direct effect on the resulting uncertainty, therefore it was carefully selected based on values observed after analysing existing data. Multiple sources of uncertainty have been examined, and this could provide the basis for a fully probabilistic uncertainty assessment framework where Monte-Carlo experiments on the input and alternative conceptualizations are performed (Purvis et al., 2008). However, combining the full parameter space in a probabilistic framework would imply prohibitive computational effort, and we feel that the present analysis remains informative. In addition, the current approach could be extended to the full geographical scale of LISCOAST (Vousdoukas et al., 2018a;Vousdoukas et al., 2018b). Finally, the selection of uncertainty sources can be deemed subjective, as with any non-exhaustive analysis (Uusitalo et al., 2015), and in the following paragraphs we try to underline the aspects which could be interesting for further investigation in future research.

The accuracy of ESL projections is affected by the atmospheric and ocean model resolution (Cavaleri and Bertotti, 2004;Calafat et al., 2014), as well as to including (or not) non-linear interactions between ESL components (Arns et al., 2015) and waves in the analysis (Serafin and Ruggiero, 2014;Vitousek et al., 2017). Previous studies have further shown that ESLs can be over-predicted if the model does not consider shoreline retreat under SLR (Du et al., 2018;Idier et al., 2017;Pickering et al., 2017), or storm-induced inundation (Bertin et al., 2014). Wahl et al. (2017) quantified the uncertainties from the PDF type used in the extreme value analysis and from the use of different ESL datasets (see also Muis et al., 2017). The present study does not include hydraulic models in the studied inundation approaches, as they are computationally expensive and complex to implement. Dynamic inundation simulations have been shown to be more reliable (Ramirez et al., 2016). However, Vousdoukas et al. (2016) have shown that VI can be a good surrogate when computational efficiency is the priority, as also demonstrated by Breilh et al. (2013). In flatter terrains VI tends to be outperformed by hydraulic models, or





other empirical approaches, such as the Flood Index Method (Dottori et al., 2016a). In terms of more robust inundation modelling, smaller-scale studies have proven the validity of models which resolve nearshore waves, erosion and dune overwash (McCall et al., 2010). Such detailed modelling, however, is not yet feasible beyond local scales due to the lack of data and computational resources.

The treatment of uncertainty in exposure is an aspect that has not been studied but that can have a strong effect on the estimated losses. This effect can be amplified for projections in coastal flood risk, given the large uncertainty in future exposure under the Shared Socioeconomic Pathways (Jiang and O'Neill, 2017;Jones and O'Neill, 2016). Likewise, the alternative vulnerability setup that was formulated is arguably limited, due to a lack of coastal flood damage data and consequent absence of coastal vulnerability studies that could produce alternative DDFs. The use of alternative risk
assessment methodologies (Hallegatte et al., 2013;Winsemius et al., 2013) can act as an additional source of epistemic uncertainty, but is not presently addressed.

The present contribution assesses multiple sources of uncertainty, some of which have been seldom studied previously. It provides insights of their relative importance in terms of their effect on the estimated losses, and can raise awareness to the coastal flood risk modelling community on the critical factors that need to be treated in future modelling attempts. For some
of the above factors, recent advances have been made to improve CFR assessment. Recently, there has been an increase in the number of studies and datasets related to future wave conditions (Fan et al., 2014;Hemer et al., 2013;Mentaschi et al., 2017), that can support large scale CFR assessments. The uncertainty related to tidal contributions can be constrained by estimating site-specific α-factor estimates (see equation 4). For example, the Monte-Carlo simulations shows that for the studied areas, confining α-factor within a range from 0.60 to 0.70 results in more realistic ESLs for all return periods (Table
4). Such reduction of the $α$ range reduces the related EAD uncertainty by nearly 50%. An alternative but more computationally expensive approach is to explore the full range of uncertainty from all ESL components, expressing them first as probability density functions (PDFs), and combining through Monte Carlo simulations in order to generate probabilistic projections of ESLs (Vousdoukas et al., 2018b).

Other critical uncertainty factors are more challenging to deal with. For example, the present findings show that the results
are strongly affected by the DEM quality, and even if high accuracy DEM data are available, the estimated CFR is very sensitive to the spatial resolution at which the analysis is carried out. However, considering coarser resolutions is inevitable for large-scale analyses and further research is needed to understand how critical this effect can be. Moreover, the present findings highlight the urgent need to generate large-scale, but high detail datasets of coastal protection standards, as the absence of such information introduces substantial uncertainty in any CFR analysis. Last but not least comes the uncertainty
related to human behaviour. Changes in exposure can be substantial under different political, social and economic settings (O'Neill et al., 2014), while vulnerability can be reduced simply as a result of societies learning to live with flood hazards (Bouwer and Jonkman, 2018).




## 6. Conclusions

The present study reports results from an analysis of epistemic uncertainty in a large-scale assessments of present and future Coastal Flood Risk (CFR). We use LISCOAST, a modular, integrated framework developed by the Joint Research Centre of the European Commission to assess the relative importance of (i) the contributions of waves and tides to Extreme Sea Levels

(ESLs); (ii) different algorithms for inundation mapping; (iii) digital elevation data (DEM); (iv) coastal flood protection information; (v) vulnerability assumptions; and (vi) different climate change projections.

The developed framework is applied in a dual case study that spans across two spatial scales and consists of: (i) a local application in Ria Formosa, Algarve, Portugal, a coastal stretch of approx. 36 km; and (ii) a regional application along the Iberian Peninsula coastline, including Portugal and Spain, with a total coastal length of 6767 km.

Digital elevation data (DEM) from SRTM are validated against LIDAR data from the regional study area, resulting in average vertical BIAS of 1.20 m and RSME of 2.15 m. We also find that reducing the DEM resolution from 10 m to 100m can change the estimated EAD by 200%, while resolution coarser than 20 m fails to resolve the dune structure, which acts as natural flood protection at the study site.

After estimating expected errors for the flood defines heights and implementing them as perturbations we find that such

uncertainty can alter the EAD within 30%-60%.

ESLs are driven by the combination of changes in the mean sea level, storm surge, tides, and waves. We find that estimated EAD can almost double after including wave setup, even though the latter is often neglected in CFR assessments. So is the spring-neap tidal variability which can alter EAD estimates between 33% and 66%.

Altering inundation approaches can result in EAD differences between 25% and 45%, however deviations are mostly related

to omitting hydrological connectivity. As the Iberian coastline is characterized by a steep terrain, the static approach with hydrological connectivity produces comparable results with a more elaborate approach which considers limitations in the effective flood water volume.

Altering the vulnerability according to the range implied by previously used depth damage functions showed to have minor contributions to the overall uncertainty. Considering baseline CFR estimation, the way wave contributions to ESLs are

25 treated, appears to be the dominant source of epistemic uncertainty at both study areas. DEM quality and resolution is highlighted as the second most important factor at the local case study.

Uncertainty in projected CFR depends on whether absolute or relative CFR changes are studied. Absolute CFR changes at the regional case study are more sensitive to wave contributions to ESLs, the quality of coastal protection information and the inundation algorithm used. All the above dominate the uncertainty of climate change and greenhouse gas emission

predictions. Relative changes in future CFR are more sensitive to the coastal protection information and the climate prediction skill, while tidal variability and greenhouse gas emissions show comparable uncertainty.



*Data availability*. This work relied entirely on public data as inputs, which are available from the providers cited in section 3. Results of the work can be downloaded from the LISCOAST data collection of the JRC data repository (http://data.jrc.ec.europa.eu/collection).

*Competing interests*. The authors declare that they have no conflict of interest.

*Acknowledgments.* The research leading to these results has received funding from the EU Seventh Framework Programme FP7/2007-2013 under grant agreement no 603864 (HELIX: "High-End cLimate Impacts and eXtremes"; www.helixclimate.eu). Alessandra Bianchi is gratefully acknowledged for her help in producing some of the maps shown in
the illustrations.

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

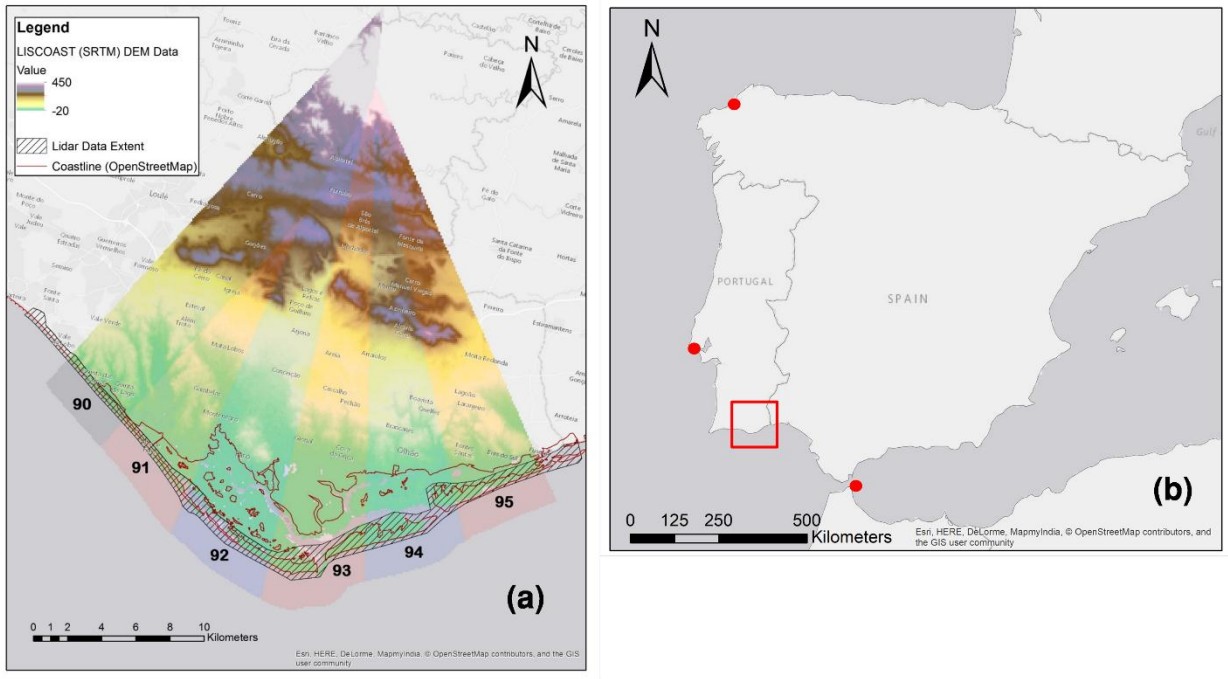

**Figure 1. Case study areas: (a) Ria Formosa, Algarve, Portugal: The alternating blue and red areas, numbered 90 to 95, show the basic coastal segments considered in LISCOAST, while the striped black overlay shows the LIDAR data extent; (b) the Iberian Peninsula, with the Ria Formosa highlighted.**





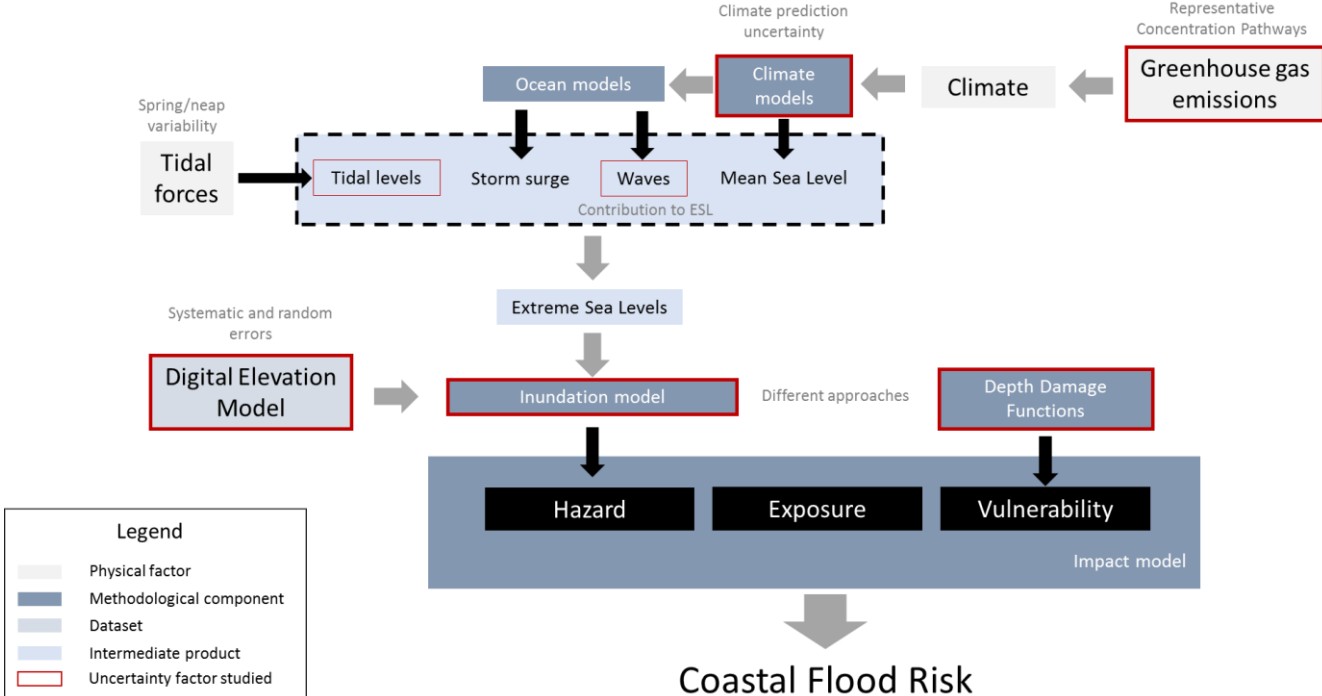

**Figure 2. The risk assessment chain of LISCOAST with the studied sources of epistemic uncertainty highlighted with red outline colour.**





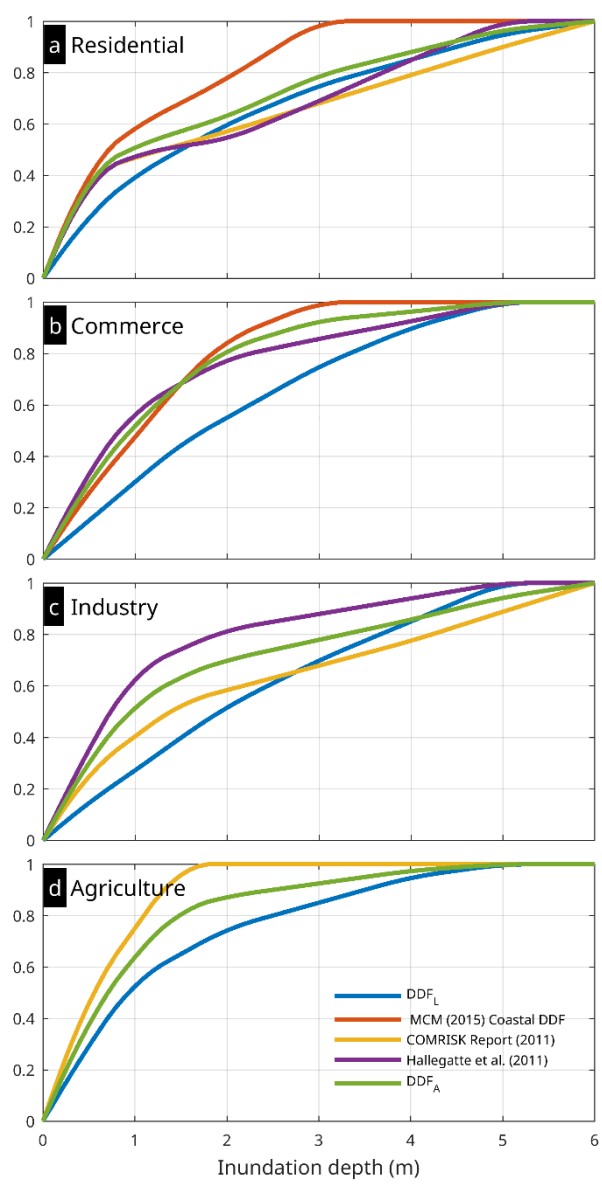

**Figure 3. Comparisons of the standard DDFs used in LISCOAST (DDF_L, averaged over Spain and Portugal) against the compiled ones from previous coastal applications (see also Table 3) and the final modified ones used for the sensitivity analysis (DDF_A). Values in the vertical axis indicate the fraction of maximum damage.**





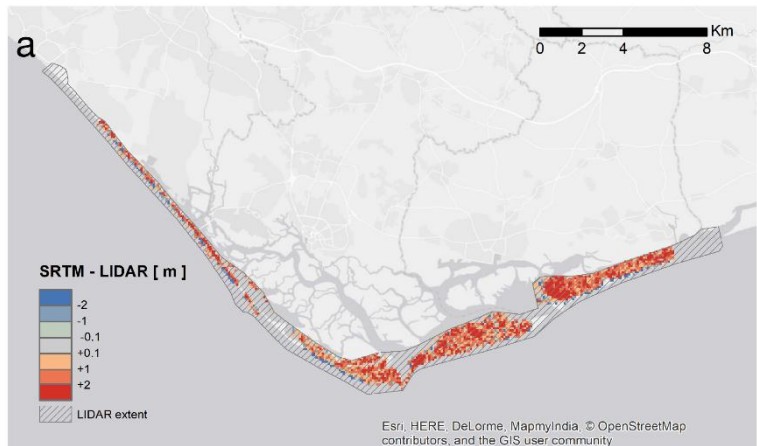

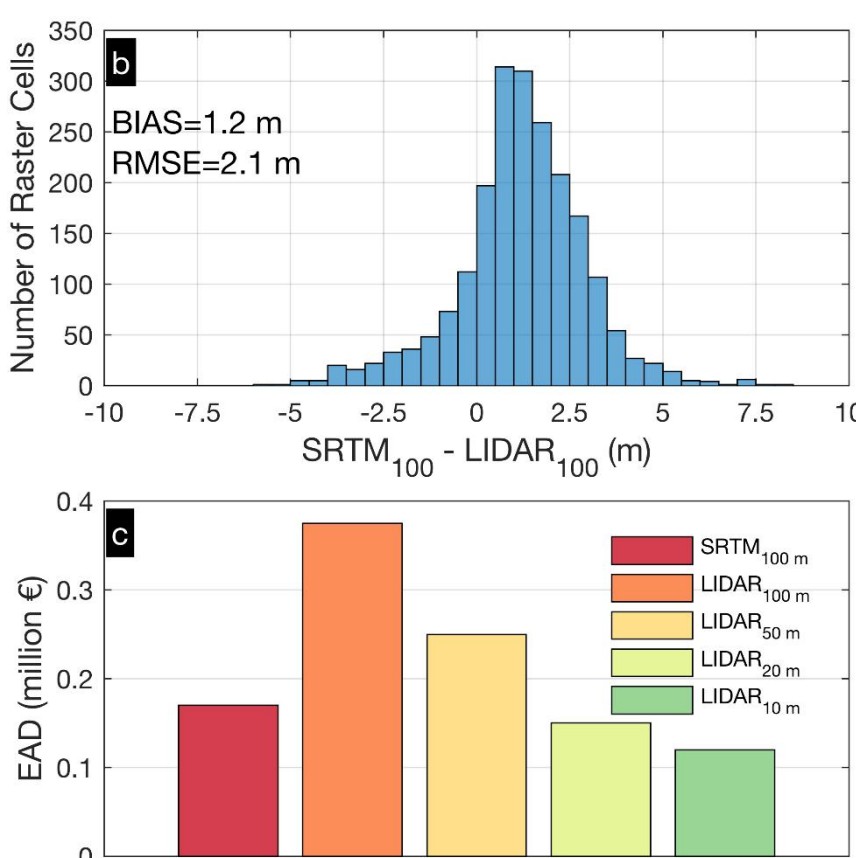

**Figure 4. Differences in elevation between the LIDAR and SRTM DEM (subsampled at 100 m) along Ria Formosa, Portugal shown in a spatial map (a), and in a histogram (b). Effect on the DEM used to estimate the baseline expected annual damage along the same area (c): the bar plot shows results for SRTM and the LIDAR DEM subsampled at 10, 20, 50 and 100 m.**



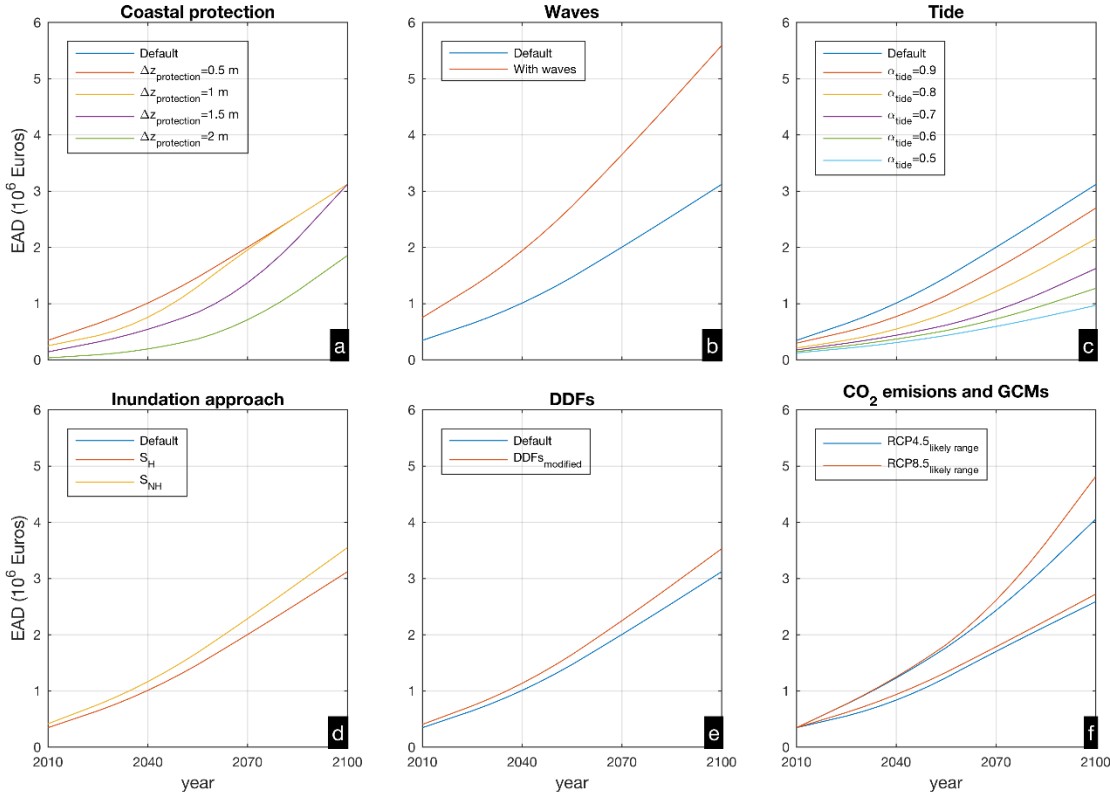

**Figure 5. Evolution of EAD during the present century for the Ria Formosa site under the different setups studied. The plots show Expected Annual Damage in million euros for different scenarios of additional coastal protection (a); including wave contributions in the Extreme Sea Levels (b); considering neap-spring tidal variability (c); applying different inundation approaches (d); and different depth damage functions (DDFs; e). All values in a-e correspond to the median case, averaged among RCP4.5 and RCP8.5. In addition the very likely range (5th-95th quantile) of both RCPs is shown (f).**





**Figure 6. Evolution of EAD during the present century for the Iberian Peninsula site under the different setups studied. The plots show Expected Annual Damage in million euros for different scenarios of additional coastal protection (a); including wave contributions in the Extreme Sea Levels (b); considering neap-spring tidal variability (c); applying different inundation approaches (d); and different depth damage functions (DDFs; e). All values in a-e correspond to the median case, averaged among RCP4.5 and RCP8.5. In addition the very likely range (5th-95th quantile) of both RCPs is shown (f).**





**Figure 7. Relative contributions to uncertainty in estimated EAD: Variability in the baseline EAD among the studied factors (a,b); the absolute (ΔESL; c,d) and relative EAD change (ΔESL%; e,f) towards the end of the century; and very likely range (g-h). Comparisons are done for the Ria Formosa (a,c,e,g) and the Iberian Peninsula (b,d,f,h) and the horizontal dashed line shows the**
5    **estimates for the *Default case* (see Section 3.2).**





## Table 1. Overview of the Case study areas.

|  | Case Number | Case Description | Case Abbrev. | Spatial Scale (Coastline Length) | Number of LISCOAST segment units |
|---|---|---|---|---|---|
| Regional (Meso) | 1 | Ria Formosa, Faro, Portugal | Faro | 36 km | 6 |
| International (Macro) | 2.a. | Portugal | PT | 1793 km | 168 |
|  | 2.b. | Spain | ES | 4964 km | 589 |

5 **Table 2. Overview of the considered climate change scenarios an return periods in the present study.**

| CLIMATE DRIVERS | | |
|---|---|---|
| EMISSION SCENARIOS | 2 | RCP 4.5, RCP 8.5 |
| CLIMATE MODELS | 6 | ACCESS1.3, EC-EARTH, CSIRO-Mk3.6.0, GFDL-CM3, HadCM3, HadGEM2-ES |
| HAZARD MODELING | | |
| RETURN PERIODS | 8 | 5, 10, 20, 50, 100, 200, 500, 1000 years |
| QUANTILES | 3 | $5^{th}$, $50^{th}$, $95^{th}$ |



**Table 3: Information on previously reported DDFs developed for coastal applications (upper panel) and land use classes which they consider (lower panel).**

| Study | Description |
|---|---|
| MCM Manual (Viavattene et al., 2015;Viavattene et al., 2018) | Residential and commercial coastal DDFs for typical UK properties. Adaptation of the fluvial DDFs with an uplift factor to account for salinity. |
| COMrisk Report (Kystdirektoratet, 2004) | Coastal DDFs for the Wadden Sea (estuarine environment) |
| Hallegate et al. (2011) | Coastal DDFs for Copenhagen |

| Study | Damage categories | | | | |
|---|---|---|---|---|---|
| | Residential | Commercial | Industry | Agriculture | Infrastructure |
| MCM Manual | x | x | | | |
| COMRISK Report | x | | x | x | |
| Hallegatte et al. (2011) | x | x | x | | |

**Table 4: Results of the Monte-Carlo analysis for the α ratio, for the three considered tide gauges.**

| | RP [years] | 5 | 10 | 20 | 50 | 100 | 200 | 500 | 1000 |
|---|---|---|---|---|---|---|---|---|---|
| α-factor values | Cascais, PT | 0.64 | 0.65 | 0.65 | 0.65 | 0.65 | 0.67 | 0.68 | 0.68 |
| | La Coruna, ES | 0.69 | 0.69 | 0.70 | 0.70 | 0.70 | 0.70 | 0.70 | 0.70 |
| | Ceuta, ES | 0.63 | 0.64 | 0.64 | 0.64 | 0.64 | 0.64 | 0.65 | 0.65 |





## Appendix



**Figure A1: Comparison of the SRTM DEM, against the LIDAR at Ria Formosa, Portugal. The comparison takes place along 12 transects (a-l), which are shown in the map of the area below and includes the SRTM at 100 m and the LIDAR data at their original resolution of 0.5 m, as well as subsampled at 5, 10, 20, 50 and 100 m.**