# Peer review of "Understanding epistemic uncertainty in large-scale coastal flood risk assessment for present and future climates"

_Natural Hazards and Earth System Sciences, 2018_

## Referee Comment (RC1) · Anonymous Referee #1 · 1 Jun 2018

This an interesting and timely manuscript that explores, in a quantitative manner, the different sources of uncertainty in large-scale coastal flood risk modelling for two case studies: a regional application for the Iberian Peninsula and a local case study in Faro, Portugal. The study concludes that uncertainty from ESL contributions, particularly the consideration of waves, exceeds the uncertainty introduced by the use of the GHG emission projections and climate models used in the analysis. Further, it reports that the information on coastal protection levels and elevation is similarly important.

The manuscript is very well written and the authors have done a very good job in clearly explaining the assumptions of the study, presenting the results and discussing their

findings. I do feel that the ms could be clearer on some methodological descriptions (I have made some suggestions below), but this is not a major comment.

I certainly recommend the paper for publication – nevertheless I think that, before publication, the authors should consider addressing a series of points that I raise below. I hope that the authors find these comments useful for improving the manuscript.

1. I find that the use of what the authors call "bathtub approach" (i.e. the static approach that does not consider connectivity) is unnecessary (and in a way incorrect) and does not add much to the manuscript. Only very few studies have applied the bathtub approach without considering hydrological connectivity in the last 10 years (or longer), as it is not correct to assume that all pixels below a certain elevation would belong to the coastal flood plain, even if they are not connected to the ocean. This would, in some locations, lead to including to the floodplain some low-elevation inland areas that can be hundreds of kilometres away from the coast. I would therefore recommend the authors to remove this method (and the respective results) from the ms, or at least limit the distance from the coastline at which such areas (pixels) are included in the calculation of the flood extent. The term bathtubo includes the connectivity consideration and I would therefore find the Snh approach obsolete.

2. I understand the need for the coastal segmentation (pg. 3, line 14) – however, I am unsure as to how the authors have implemented it and specifically how they have defined the inland boundaries of the segments (perpendicular to the coast?); and how they have addressed the problem of "spill-over" of water between the segments. I would assume that this problem could substantially affect the results of the VI method as water does not stop at the inland boundaries but rather propagates to neighbouring segments. I suggest that the authors add some lines to the manuscript (or in the supplementary material) providing some additional information on this process.

3. The analysis presented in the manuscript is based on the LISCOAST framework. However, this framework is not described in the manuscript and the reference (which

has been accepted for publication) is not yet available; some basic information on LISCOAST would therefore be useful to include in the main text or as supplementary material.

4. To address the assumption that all extreme events coincide with high tide the authors use a modulation factor. If I understand correctly, this factor only considers the spring-neap tide variability, thus not accounting for the actual variability of the tide during a storm. In the case of the VI method this could lead to substantial overestimation of the volume of water (particularly in the Atlantic coast of the Iberian peninsula) as it assumes that there is always high water. It would be useful to clearly mention and discuss this point.

5. If I understand correctly from Fig. 1b the tide gauge used for the Mediterranean coast of the Iberian Peninsula is not located in the Iberian Peninsula (seems to be in Africa, on Spanish territory) and may not be the most representative one for the Mediterranean coast as it is located next to the strait and could be affected by currents? Why didn't the authors use other tide gauges from e.g. Barcelona or Valencia? In the same figure, for the sake of completeness, it might also be useful to include in the caption that red dots represent tide gauges.

6. Based on the figures (e.g. fig.4) it seems to me that the highest differences between SRTM and Lidar appear along the barrier islands, mostly in areas that is actually water and where the two datasets do not perfectly overlap. In this context, it might well be that the reported bias and error are overestimated, as the water and the areas of overlap could (should?) be easily masked out. I am not necessarily suggesting that the authors should repeat the calculation but if what I am suggesting is correct, they should discuss this point in the manuscript as I assume that a comparison of the two DEMs should include masking out the water surfaces (while also ensuring that the extent of the two datasets is exactly the same, i.e. that they are co-registered and overlap "completely").

7. Two minor comments about the figures: In figure 5 the default lines cannot be clearly

seen as they coincide with others. Also, figure A1 leaves the impression that the SRTM includes non-integer values (e.g. A1f). Might there be something wrong there, or have the authors performed some type of interpolation, or did they use one of the newer versions which include non-integer values? If the latter is correct, they should cite correctly which version they used. Also, the x-axis is missing numbering

8. A suggestion - I do not want to be pedantic but "large scale" is actually "small area" or local (because 1:10 scale is larger than 1:100). I am aware that the term is widely used in the context that the authors use it but I would suggest to change this to e.g. "broad scale", "global scale" or "large area" as suggested in some other journal.

---

## Referee Comment (RC2) · Anonymous Referee #2 · 12 Jun 2018

The manuscripts quantifies different types of epistemic uncertainties and the effects they can have on the results from broad-scale flood risk assessments. Here, the authors focus on two case studies, one relatively small (which I wouldn't necessarily refer to as "large-scale") and the other one much larger, covering the Iberian coast. Uncertainties are assessed for most of the key variables involved in flood risk assessments. I find the manuscript really interesting and well written. The analysis is technically sound using the latest data sets and the conclusions are supported by the results. I only have some minor comments which I would like to see addressed in a revised version.

1-26 and elsewhere: check order of references
2-14 Was there a specific reason for using 25 km?

2-25 I didn't understand the reference to "ten years into the present century" in the context of the sentence.

4-13 This approach seems a bit outdated and could be removed without losing any relevant information.

5-23 It wasn't clear to me what kind of tidal signal is used here and where it comes from. Is the tide level assumed constant throughout an event?

6-7 I am wondering what the predominant type of protection is in the study areas and whether or not it could make sense to actually try and calculate runup (e.g. with the Stockdon formula) for places where dunes are the primary defense and where erosion (and possibly breaches) are initiated with overtopping. I am not saying the authors need to change their approach, but interested to hear their thoughts on this.

7-1 I think this needs a reference.

10-10 "in large-scale..."

12-14/15 This sentence needs rewording, I didn't understand what it is telling me.

Fig 3 Not all curves are visible in each panels, do they overlap? Maybe consider using dashed lines for some of them.

Fig A1 Needs numbers on the x-axis.

NHESSD

---

## Author Comment (AC1) · 3 Jul 2018

This an interesting and timely manuscript that explores, in a quantitative manner, the different sources of uncertainty in large-scale coastal flood risk modelling for two case studies: a regional application for the Iberian Peninsula and a local case study in Faro, Portugal. The study concludes that uncertainty from ESL contributions, particularly the consideration of waves, exceeds the uncertainty introduced by the use of the GHG emission projections and climate models used in the analysis. Further, it reports that the information on coastal protection levels and elevation is similarly important. The manuscript is very well written and the authors have done a very good job in clearly

explaining the assumptions of the study, presenting the results and discussing their findings. I do feel that the ms could be clearer on some methodological descriptions (I have made some suggestions below), but this is not a major comment. I certainly recommend the paper for publication – nevertheless I think that, before publication, the authors should consider addressing a series of points that I raise below. I hope that the authors find these comments useful for improving the manuscript.

Authors: we would like to thank the reviewer for his/her positive feedback and the constructive comments, which we did our best to address.

1. I find that the use of what the authors call "bathtub approach" (i.e. the static approach that does not consider connectivity) is unnecessary (and in a way incorrect) and does not add much to the manuscript. Only very few studies have applied the bathtub approach without considering hydrological connectivity in the last 10 years (or longer), as it is not correct to assume that all pixels below a certain elevation would belong to the coastal flood plain, even if they are not connected to the ocean. This would, in some locations, lead to including to the floodplain some low-elevation inland areas that can be hundreds of kilometres away from the coast. I would therefore recommend the authors to remove this method (and the respective results) from the ms, or at least limit the distance from the coastline at which such areas (pixels) are included in the calculation of the flood extent. The term bathtub includes the connectivity consideration and I would therefore find the Snh approach obsolete.

Authors: Following also a suggestion from reviewer 2 we have removed Snh from the revised version.

2. I understand the need for the coastal segmentation (pg. 3, line 14) – however, I am unsure as to how the authors have implemented it and specifically how they have defined the inland boundaries of the segments (perpendicular to the coast?); and how they have addressed the problem of "spill-over" of water between the segments. I would assume that this problem could substantially affect the results of the VI method
as water does not stop at the inland boundaries but rather propagates to neighbouring segments. I suggest that the authors add some lines to the manuscript (or in the supplementary material) providing some additional information on this process.

Authors: This is a valid point on this common issue with the inevitable use of segments. Our solution is to consider segments for the inundation modeling which extend beyond the ones defined for the impact assessment. This implies that the impact modeling segments do not overlap but the inundation modelling segments do and then are combined always considering the highest flood depth, before proceeding to the impact assessment.

3. The analysis presented in the manuscript is based on the LISCOAST framework. However, this framework is not described in the manuscript and the reference (which has been accepted for publication) is not yet available; some basic information on LISCOAST would therefore be useful to include in the main text or as supplementary material.

Authors: LISCOAST is described in detail in our Nature Climate Change paper which has been accepted and will be published in the beginning of July. As this is way before the earliest possible publication date of the present study we believe that duplicating that information would not make sense. This is especially given the fact that in the present paper the approach followed is sufficiently described so that the reader can follow the research done. Still we are open to provide more details if the reviewer or the editor insists.

4. To address the assumption that all extreme events coincide with high tide the authors use a modulation factor. If I understand correctly, this factor only considers the spring neap tide variability, thus not accounting for the actual variability of the tide during a storm. In the case of the VI method this could lead to substantial overestimation of the volume of water (particularly in the Atlantic coast of the Iberian peninsula) as it assumes that there is always high water. It would be useful to clearly mention and
**discuss this point.**

Authors: This is another valid point from the reviewer and we have added a discussion in the revision. It is true that we consider the temporal evolution of only the meteorological tide during the event and not of the astronomical one. Given the coupling between tide and extreme weather is stochastic we cannot predict on which phase of the daily tidal cycle the storm will occur, so we made the assumption of constant tidal level. This in effect implies that the peaks of the meteorological and astronomic tide will coincide, a factor which can potentially result in overestimations. A secondary overestimation factor is the fact that the tide remains constant during the whole event, but this is minor since most of the flooding takes place during the short period that the hydrograph is near its peak. Such issues are the result of the transition from static to more dynamic inundation approaches and a dedicated paragraph has been added in the discussion of the revision.

5. If I understand correctly from Fig. 1b the tide gauge used for the Mediterranean coast of the Iberian Peninsula is not located in the Iberian Peninsula (seems to be in Africa, on Spanish territory) and may not be the most representative one for the Mediterranean coast as it is located next to the strait and could be affected by currents? Why didn't the authors use other tide gauges from e.g. Barcelona or Valencia? In the same figure, for the sake of completeness, it might also be useful to include in the caption that red dots represent tide gauges.

Authors: The comment is fair but the tide gauge selection was an inevitable artefact of data availability during the time of the study. The Ceuta station was the closest to the Mediterranean Spanish coast, among the Spanish tide gauges available from the UHSLC dataset (see also:Âăhttps://uhslc.soest.hawaii.edu/datainfo/). For example the tide gauges from Barcelona or Valencia suggested by the reviewer were not available in the UHSLC dataset. While some contamination from Atlantic tidal currents is likely, we believe that it does not have a discernible effect on the analysis.Âă
6. Based on the figures (e.g. fig.4) it seems to me that the highest differences between SRTM and Lidar appear along the barrier islands, mostly in areas that is actually water and where the two datasets do not perfectly overlap. In this context, it might well be that the reported bias and error are overestimated, as the water and the areas of overlap could (should?) be easily masked out. I am not necessarily suggesting that the authors should repeat the calculation but if what I am suggesting is correct, they should discuss this point in the manuscript as I assume that a comparison of the two DEMs should include masking out the water surfaces (while also ensuring that the extent of the two datasets is exactly the same, i.e. that they are co-registered and overlap "completely").

Authors: The LIDAR data from Ria Formosa include only sub-aerial areas as the sensor used could not collect bathymetric data. So both datasets exclude the submerged areas, and the same applies for the patches shown Figure 4. Maybe the maps gave the reviewer the impression that submerged parts are included, because of the rather complex Ria Formosa topography.Âă Moreover, as it can be seen from the coastal profile comparison of Figure A1, the difference could be attributed to the fact that nearshore SRTM cells have, in many cases, values that converge faster to zero compared to the LIDAR values for cells close to or adjacent to the sea. This in turn could be attributed to the aggregation/interpolation process used within the SRTM dataset. In any case, all necessary steps have been taken to ensure the complete alignment and overlap of the two datasets.Âă

7. Two minor comments about the figures: In figure 5 the default lines cannot be clearly seen as they coincide with others. Also, figure A1 leaves the impression that the SRTM includes non-integer values (e.g. A1f). Might there be something wrong there, or have the authors performed some type of interpolation, or did they use one of the newer versions which include non-integer values? If the latter is correct, they should cite correctly which version they used. Also, the x-axis is missing numbering

Authors: We have improved all figures according to both reviewers' suggestions. We have added dashed lines in Fig 5 and x ticks in Fig A1. As the reviewer correctly
guesses, the decimal number occur from interpolations being part of the resampling process from the native SRTM scale to the scale needed by LISCOAST cells (100.0 m x 100.0 m).

8. A suggestion - I do not want to be pedantic but "large scale" is actually "small area" or local (because 1:10 scale is larger than 1:100). I am aware that the term is widely used in the context that the authors use it but I would suggest to change this to e.g. "broad scale", "global scale" or "large area" as suggested in some other journal.

Authors: We understand the reviewer's point but we hope that he/she agrees that this is not an important issue, as long as 'large-scale' is properly defined in the beginning of the paper. In our case we clearly state that the work focusses on continental and global scale analyses and if we switch from 'large' to 'global' we are concerned that we will be excluding 'continental' which is also a scale considered by LISCOAST. Also large-scale is a term commonly used in literature and for that reason we decided to keep it. Still we will be happy to change in case the editor and the reviewer insist.

---

## Author Comment (AC2) · 3 Jul 2018

The manuscripts quantifies different types of epistemic uncertainties and the effects they can have on the results from broad-scale flood risk assessments. Here, the authors focus on two case studies, one relatively small (which I wouldn't necessarily refer to as "large-scale") and the other one much larger, covering the Iberian coast. Uncertainties are assessed for most of the key variables involved in flood risk assessments. I find the manuscript really interesting and well written. The analysis is technically sound using the latest data sets and the conclusions are supported by the results. I only have some minor comments which I would like to see addressed in a revised version. Authors: we would like to thank the reviewer for his positive feedback and the constructive comments, which we did our best to address. In our opinion none of our case studies are really 'large-scale', but they serve for validation and sensitivity analysis of a methodology applied at continental and global scale.

1-26 and elsewhere: check order of references

Authors: All references are imported using reference manager software using the journal's template which means that they should be included properly. In case this is not true we will correct accordingly.

2-14 Was there a specific reason for using 25 km?

Authors: The decision to do the analysis in 25 km was considered as a good compromise, as it allows for (i) acceptable calculation times for the flood simulations (which increase exponentially with the size of the domain; especially when the model Lisflood is used (Vousdoukas et al., 2018;Vousdoukas et al., 2016)) and sufficient resolution in the impact results for large scale standards (considering that previous global analyses are at 100 m). It is also important to highlight that the principal analysis takes place at 100 m.

2-25 I didn't understand the reference to "ten years into the present century" in the context of the sentence.

Authors: We added 'every' before the sentence and we hope it now reads better.

4-13 This approach seems a bit outdated and could be removed without losing any relevant information.

Authors: Following both reviewers' suggestions we have excluded Snh from the revision.

5-23 It wasn't clear to me what kind of tidal signal is used here and where it comes from. Is the tide level assumed constant throughout an event?
Authors: This comment is relevant to comment 4 from reviewer 1 and we hope it has been made clear in the revision.

6-7 I am wondering what the predominant type of protection is in the study areas and whether or not it could make sense to actually try and calculate runup (e.g. with the Stockdon formula) for places where dunes are the primary defense and where erosion (and possibly breaches) are initiated with overtopping. I am not saying the authors need to change their approach, but interested to hear their thoughts on this.

Authors: The issue of considering run up is recurring in most of our papers and has also been discussed in some of them. The use of the generic approximation for wave setup comes as an inevitable simplification given the absence of data on the beach face slope needed to estimate wave runup. In terms of hydrodynamics we have all the data required to apply a more elaborate method, and the leading author has substantial experience on swash zone processes(Vousdoukas. 2014:Vousdoukas et al., 2014:Vousdoukas et al., 2013:Almeida et al., 2012:Vousdoukas et al., 2012a; Vousdoukas, 2012; Schimmels et al., 2012; Vousdoukas et al., 2012b;Vousdoukas et al., 2012c;Vousdoukas et al., 2011;Vousdoukas et al., 2009), however essential topographic data are missing. In the absence of the latter we have applied the generic approximation for wave setup as we also find that it is more compatible with the current state of the art in flood and risk assessments. Most studies apply the static inundation approach which anyway overestimates flood extents (Vousdoukas et al., 2016). The wave runup height is not persistent during an extreme event and its use would result in further overestimation. Even in studies which apply hydrological models for coastal inundation(Ramirez et al., 2016), the time steps of the simulations are not small enough to resolve wave oscillations and wave runup would be an overprediction of the forcing water level. For that reason wave setup, being a slower, more persistent episodic elevation of the sea level than wave runup, was chosen.

7-1 I think this needs a reference.
Authors: References have been added.

10-10 "in large-scale: : :"

Authors: The text has been corrected accordingly.

12-14/15 This sentence needs rewording, I didn't understand what it is telling me.

Authors: The reviewer is right and the text has been corrected accordingly.

Fig 3 Not all curves are visible in each panels, do they overlap? Maybe consider using dashed lines for some of them.

Authors: In line also with comment 7 from reviewer 1 we have added dashed lines.

Fig A1 Needs numbers on the x-axis.

Authors: In line also with comment 7 from reviewer 1 we have tick labels in Fig A1. References Almeida, L. P., Vousdoukas, M. I., Ferreira, Ó., Rodrigues, B. A., and Matias, A.: Thresholds for storm impacts on an exposed sandy coastal area in southern Portugal, Geomorphology, 143-144, 3-12, 10.1016/j.geomorph.2011.04.047, 2012. Ramirez, J. A., Lichter, M., Coulthard, T. J., and Skinner, C.: Hyper-resolution mapping of regional storm surge and tide flooding: comparison of static and dynamic models, Nat. Hazards, 82, 571-590, 10.1007/s11069-016-2198-z, 2016. Schimmels, S., Vousdoukas, M. I., Oumeraci, H., and Wziatek, D.: Wave run-up observations at revetments with different porosity, 33rd International Conference on Coastal Engineering, Santander, Spain, July 1-6, 2012, 2012. Vousdoukas, M. I., Velegrakis, A. F., Dimou, K., Zervakis, V., and Conley, D. C.: Wave run-up observations in microtidal, sedimentstarved pocket beaches of the Eastern Mediterranean, Journal of Marine Systems, 78, S37-S47, 2009. Vousdoukas, M. I., Almeida, L. P., and Ferreira, O.: Modelling storm-induced beach morphological change in a meso-tidal, reflective beach using XBeach, J. Coast. Res., 1916-1920, 2011. Vousdoukas, M. I.: Erosion/accretion and multiple beach cusp systems on a meso-tidal, steeply-sloping beach, Geomorphology, 141-142, 34-46, doi:10.1016/j.geomorph.2011.12.003, 2012. Vousdoukas, M. I.,

NHESSD
Almeida, L. P., and Ferreira, Ó.: Beach erosion and recovery during consecutive storms at a steep-sloping, meso-tidal beach, Earth Surf. Processes Landforms, 37, 583-691, 10.1002/esp.2264, 2012a. Vousdoukas, M. I., Ferreira, O., Almeida, L. P., and Pacheco, A.: Toward reliable storm-hazard forecasts: XBeach calibration and its potential application in an operational early-warning system, Ocean Dyn., 62, 1001-1015, 10.1007/s10236-012-0544-6, 2012b. Vousdoukas, M. I., Wziatek, D., and Almeida, L. P.: Coastal vulnerability assessment based on video wave run-up observations at a mesotidal, steep-sloped beach, Ocean Dyn., 62, 123-137, 10.1007/s10236-011-0480x, 2012c. Vousdoukas, M. I., Wachler, B., Almeida, L. P., Ferreira, O., Alexandrakis, G., Velegrakis, A. F., and Schimmels, S.: Predicting beach-face rotation on a mesotidal, steeply sloping, beach, 7th International Conference on Coastal Dynamics, Bordeaux, France, 24-28 June, 2013. Vousdoukas, M. I.: Observations of wave runup and groundwater seepage line motions on a reflective-to-intermediate, meso-tidal beach, Mar. Geol., 350, 52-70. http://dx.doi.org/10.1016/j.margeo.2014.02.005, 2014. Vousdoukas, M. I., Kirupakaramoorthy, T., Oumeraci, H., de la Torre, M., Wübbold, F., Wagner, B., and Schimmels, S.: The role of combined laser scanning and video techniques in monitoring wave-by-wave swash zone processes, Coastal Eng., 83, 150-165, http://dx.doi.org/10.1016/j.coastaleng.2013.10.013, 2014. Vousdoukas, M. I., Voukouvalas, E., Mentaschi, L., Dottori, F., Giardino, A., Bouziotas, D., Bianchi, A., Salamon, P., and Feyen, L.: Developments in large-scale coastal flood hazard mapping, Natural Hazards and Earth System Science, 16, 1841-1853, 10.5194/nhess-16-1841-2016, 2016. Vousdoukas, M. I., Mentaschi, L., Voukouvalas, E., Alessandra, B., Francesco, D., and Feyen, L.: Climatic and socioeconomic controls of future coastal flood risk in Europe Nature Climate Change, accepted, 2018.

---

## Author Response (AR1)

We would like to thank the reviewers and the editor for their positive and constructive comments. We are submitting a revised version according to all the raised points:

Following a suggestion from both reviewers we have removed the Snh inundation approach (static without hydrological connection) from the revised version and we have changed the text and figures accordingly.

We clarify the point from Reviewer 1 related to the segment definition and the management of "spill-over" of water issues between the segments.

We clarify the first reviewer's concerns regarding the contributions of tides to ESLs.

We have improved figures 5 and A1 according to both reviewers' suggestions.

We applied other minor corrections requested and we have updated references which were published during the discussion period.

[revised manuscript text omitted]